# Technical note: Evaluation of three machine learning models for surface ocean CO2 mapping

Jiye Zeng[1], Tsuneo Matsunaga[1], Nobuko Saigusa[1], Tomoko Shirai[1], Shin-ichiro Nakaoka[1], Zheng-Hong Tan[2]

[1]National Institute for Environmental Studies, Tsukuba, Japan
[2]Department of Environmental Science, Hainan University, China

*Correspondence to*: Jiye Zeng (zeng@nies.go.jp)

**Abstract.** Reconstructing surface ocean $CO_2$ from scarce measurements plays an important role in estimating oceanic $CO_2$ uptake. There are varying degrees of differences among the 14 models included in the Surface Ocean $CO_2$ Mapping (SOCOM) inter-comparison initiative, in which five models used neural networks. This investigation evaluates two neural networks used in SOCOM, self-organization map and feedforward neural network, and introduces a machine learning model called support vector machine for ocean $CO_2$ mapping. The technique note provides a practical guide to selecting the models.

## 1 Introduction

The global ocean is a major sink for anthropogenic carbon and therefore an important contributor for slowing down the human-induced global warming (Stocker et al., 2013). For calculating the oceanic $CO_2$ uptake, various models have been used to interpolate scarce $CO_2$ measurements in the surface ocean spatially and temporarily to obtain basin-wide (e.g. Zeng et al., 2002; Lefevre et al., 2005; Chierici et al., 2006; Sarma et al., 2006; Jamet et al., 2007; Friedrich and Oschlies, 2009; Telszewski et al., 2009; Takamura et al., 2010; Landschützer et al., 2013; Nakaoka et al., 2013; Iida et al., 2015; Goddijn-Murphy et al, 2015) and global ocean $CO_2$ maps (Takahashi et al., 2002, 2009 and 2014; Park et al., 2010. Rödenbeck et al., 2013; Sasse et al., 2013; Jones et al., 2015; Zeng et al., 2015). The Surface Ocean $CO_2$ Mapping (SOCOM) inter-comparison initiative revealed varying degrees of differences among 14 models (Rödenbeck et al., 2015), of which 5 used neural networks. They include self-organizing maps (SOM) and feedforward neural networks (FNN). The SOM has a long history in $CO_2$ mapping (Lefevre et al., 2005; Friedrich and Oschlies, 2009; Telszewski et al., 2009; Nakaoka et al., 2013). Recently, the FNN is gaining popularity in this field (Landschützer et al., 2015; Zeng et al., 2014 and 2015). In this investigation we introduce a machine learning model called support vector machine (SVM) for ocean $CO_2$ mapping and compare the SVM with the SOM and FNN. We intend to provide a practical guide for using these machine learning models.

**2 Model Equations**

The machine learning models included in this study cannot directly model the long-term trend of $CO_2$. Therefore, we express the dependence of $CO_2$ fugacity ($fCO_2$) on year ($YR$), month ($MON$), latitude ($LAT$), and longitude ($LON$) as the sum of a nonlinear static component and a linear trend component:

$$fCO_2 = F_{static}(LAT, LON, MON) + F_{trend}(YR). \tag{1}$$

As available observations are scarce with respect to the biogeochemical properties of the surface ocean, we used sea surface temperature ($SST$), sea surface salinity ($SSS$), chlorophyll-a concentration ($CHL$), and mixed layer depth ($MLD$) as the proxy variables of space and time. These proxy variables were commonly used by models included in the SOCOM. The model equation becomes

$$fCO_2 = F_{static}(LAT, SST, SSS, CHL, MLD, dSST) + F_{trend}(YR), \tag{2}$$

where d$SST$ denotes the difference between the monthly and annual means of $SST$. Here we excluded $LON$ and $MON$. They have a circular property and therefore cannot be used directly. For instance, longitude -180 degree is geographically connected to longitude 180 degree, but numerically they appear to be two extreme longitude values to the models. Zeng et al. (2014 and 2015) circumvented this problem by using sine and cosine transformed components. Their approach could unintentionally enhance the influence of $LON$ and $MON$ on $fCO_2$ as one more derived variable from each of them were added to the model. We excluded $LON$ for the belief that the combination of $SST$, $SSS$, $CHL$, and $MLD$ contains sufficient spatial information, but retained $LAT$ for its different seasonal and geophysical meanings in the northern and southern hemispheres. Replacing $MON$ by d$SST$ also improves expressing the effect of season geographically.

**3 Data**

We extracted monthly $fCO_2$ from the track-gridded database of the Surface Ocean $CO_2$ Atlas (SOCAT) version 3.0[1] (Pfeil et al., 2013; Sabine et al., 2013; Bakker et al. 2013). The database has a 1°×1° spatial resolution and includes global measurements from 1970 to 2014. Similar to Zeng et al. (2014), we excluded some data points by these criteria: (i) $fCO_2$ values smaller than 250 µatm or larger than 550 µatm, (ii) ocean depth smaller than 500 m, (iii) salinity smaller than 25.0, and (iv) year before 1990. A total of 158,052 data points were extracted with these conditions.

---

[1] http://www.socat.info/

The monthly SST data of 1990 to 2015 were extracted from the Optimum Interpolation (OI) V2 product[2] of NOAA (Reynolds et al., 2002). The monthly SSS climatology was extracted from the World Ocean Atlas 2013 (WOA13) product[3] (Boyer et al., 2013), which contains the monthly mean SSS from June 27, 1896 to December 25, 2012. The monthly CHL climatology was calculated using the MODIS Aqua and SeaWiFS climatology[4], which covers the period of 2012 to 2015. The mean of the two CHLs was used as the CHL climatology. The mixed layer data were derived from the Monthly Isopycnal and Mixed-layer Ocean Climatology[5] of NOAA (Schmidtko et al., 2013), which includes the period of 1955 to 2012.

## 4 Machine Learning Models

The Appendix and Table 1 summarize the algorithms of the three models. Here we focus on discussing their usage in $CO_2$ mapping.

The trend in Eq.(2) cannot be modelled directly by the models. One approach to deal with the problem is to normalize the measurements to a reference year using a global rate and only model the nonlinear component. Zeng et. al. (2014) presented a method to model the linear component in Eq.(2). Instead of repeating the process, we used their annual rate of 1.5 µatm to remove trend from $fCO_2$ to normalize it to the reference year 2005, i.e.,

$$fCO_2^{normalized} = fCO_2 - 1.5 * (YR - 2005) \qquad (3)$$

Although Takahashi et al (2014) obtained a global mean rate of 1.9 µatm yr$^{-1}$, we used 1.5 µatm yr$^{-1}$ as this rate was obtained by using the gridded $fCO_2$ of SOCAT version 2. The normalized $fCO_2$ was used to model the nonlinear component in Eq.(2). In later discussions, $fCO_2$ means the normalized $fCO_2$ unless explicitly stated. Similarly, we applied the log transform of Zeng et. al. (2014) to *CHL* prior to data scaling discussed below, i.e.,

$$CHL = \log_{10}(1.0 + CHL). \qquad (4)$$

### 4.1 SMV

For a given dataset, the SVM requires a prior step to find the optimal value for the parameter $\sigma$ in Eq.(A10) and the parameter $\gamma$ in Eq.(A11). To shorten the training time, we randomly chose 10% of the measurement data in this step and obtained 0.06

---

[2] http://www.esrl.noaa.gov/psd/data/gridded/data.noaa.oisst.v2.html

[3] https://www.nodc.noaa.gov/OC5/woa13/

[4] http://oceancolor.gsfc.nasa.gov/cgi/l3

[5] http://www.pmel.noaa.gov/mimoc/

for $\sigma$ and 10 for $\gamma$. Note that these values are dependent on data scaling, which is necessary in this case to avoid overflow problem in solving Eq.(A18). We scaled all input variables LAT, SST, SSS, CHL, MLD, and dSST by their minimum and maximum to confine them in the range (0, 1), i.e.,

$$v = \frac{v - v_{min}}{v_{max} - v_{min}}. \tag{5}$$

### 4.2 FNN

Data scaling is not necessary for the FNN, but can improve its performance. Following Zeng et al. (2014), we scaled the input variables by their mean and standard deviation as

$$v = \frac{v - \bar{v}}{s}. \tag{6}$$

The output variable $f$CO$_2$ is scaled by

$$v = 0.1 + 0.8 \frac{v - v_{min}}{v_{max} - v_{min}}. \tag{7}$$

This confines the scaled $f$CO$_2$ between 0.1 and 0.9 for better response to changes of input variables. The kernel function Eq.(A4) has the property that for any input in $(-\infty, +\infty)$, the output varies between 0 and 1. For $f$CO$_2$ close to 0 or 1, a small

change in $f$CO$_2$ requires very large adjustment of model parameters, which slows down the convergence of training.

We used 64 hidden neurons for the FNN as Zeng et al. (2014) did. The learning rate in Eq.(A6) was set to 0.25 by trial-and-error. A small value makes training slow; whereas a large value may make a training diverge. The constant in Eq.(A8) was determined dynamically in each iterative training loop. It was taken as 10 times the mean of absolute differences between

modelled and observed $f$CO$_2$. We experienced that this method improves the performance of training.

### 4.3 SOM

Data scaling is critical for the SOM, as the distance defined by Eq.(A1) would be affected by variable units. We used Eq.(6) to scale input variables in training the SOM. Based on our preliminary correlation analysis, we applied a factor of 2 to enhance

the influence of *SST* and *CHL* on the distance. Using such a subjective factor is the only way to make the correlations between the output and the input variables more in line with observed correlations.

From the labelling procedure of SOM described in the Appendix, it is not difficult to see that the number of neuron cells in SOM affects the labelling and hence the prediction. Unfortunately, there is no guideline for choosing the size. Based on

previous studies (Telszewski et al., 2009 and Nakaoka et al. 2013), we used 20,000 neuron cells, roughly one neuron cell for one 1x1 grid cell of sampled areas.

**5 Model Validation**

We examined the goodness of fit by randomly selecting 10% to 50% of the data points to train the FNN and SVM, and to label the SOM; and then calculated the correlation coefficient between modelled and observed $CO_2$ of the selected data points.

The SOM yields the best correlation in the case of 10% of randomly selected data points and the correlation decreases with the number of data points (Fig.1). The reason is that for a given number of neuron cells, the fewer the data points, the less possible a neuron cell will be labelled by multiple measurements and the more likely that the prediction will find the same $CO_2$ value used for labelling. Therefore, the goodness of fit does not necessary mean good SOM modelling.

The correlations obtained by the SVM and FNN do not vary much with the number of data points. While the SVM's correlation decreases monotonically, even though by only a little, with the number of data points, the FNN's correlation obtained with 75000 data points is larger than that with 60000 data points. The FNN is known for not being able to find the global optimum in training. This case could be an indication of an imperfect training. The FNN appears inferior to SVM in all case. However, imperfect training does not account for all the differences. If we use the number of model parameters to be determined by the training as the indicator of the dimension of the model space, the FNN's dimension is far smaller than that of the SVM. The
former is determined by the number of hidden neurons and input variables, whereas the latter is determined by the number of training data. For 6 input variables, 15000 training data, and 64 hidden neurons, the number of parameters is 509 for the FNN and 15001 for the SVM.

A better indicator for the performance of the models would be the goodness of prediction. To emulate the situation that the sampled area was only a small portion of the global ocean, we evaluated the goodness of prediction by training FNN and SVM and labelling SOM with 10% of randomly selected data to make prediction for the rest of the data. Fig.2 shows that the SVM yielded the best correlation ($R^2$=0.72), the FNN fell behind ($R^2$=0.67), and the SOM performed the worst ($R^2$=0.54). The differences between predicted and observed $f$CO$_2$ are 0.1±17.4 µatm for SVM, 0.1±18.9 µatm for FNN, and 0.2±23.3 µatm
for SOM. Comparing to the variation of $f$CO$_2$ measurements, these differences are small and their uncertainties are in the same order of magnitude as the variation of measurements. Let's examine the standard deviation (STD) of $f$CO$_2$ in those grids having at least 3 data points. The track-gridded $f$CO$_2$ in SOCAT version 3.0 includes STD ranging from 0.1 µatm to 71.2 µatm and the mean is 5.2 µatm. Calculating the STD of normalized $f$CO$_2$ in the same grids and in the same months of all years yielded a mean of 12.5 µatm in the range of 0.1 µatm to 103.1 µatm. The normalization had little effect on the STD as the calculation
for non-normalized $f$CO$_2$ gives a mean STD of 14.6 µatm in the range of 0.1 µatm to 107.5 µatm.

From the algorithm of SOM in the Appendix, it is not difficult to see that the SOM does not make extrapolation – the model always approximates new inputs by the measurements used for training and approximates $f$CO$_2$ by the measurements used for

labelling; therefore, the predicted $f$CO$_2$ values are within the observed $f$CO$_2$ range (Figure 2a). Figure 2c shows that the extrapolated $f$CO$_2$ by the SVM, if any, did not exceed the observed range. To investigate the extrapolation risk, we used 200,000 data points randomly generated for SST, dSST, SSS, MLD, and CHL in the range of (0 °C, 40 °C), (-20 °C, 20 °C), (20, 50), (1 m, 1500 m), and (0 log(mg m$^{-3}$), 2 log(mg m$^{-3}$)) respectively. These ranges are larger than the corresponding observed ranges of (0 °C, 34 °C), (-13 °C, 16 °C), (24, 40), (1 m, 1000 m), and (0 log(mg m$^{-3}$),1.2 log(mg m$^{-3}$)). The SVM and FNN produced $f$CO$_2$ in the range of (267 μatm, 468 μatm) and (199 μatm, 596 μatm) respectively for the simulated samples. Comparing to the observed $f$CO$_2$ range of (240 μatm, 560 μatm), the experiment indicates that the over-extrapolation risk of the SVM is low.

**6 Differences**

Figure 3 shows $f$CO$_2$ maps in February and July, 2005, which is the reference year for normalization. In the mapping, we randomly selected 50% of the data to train the FNN and SVM and to label the SOM. All models captured the major features of observed $f$CO$_2$ distribution. The SOM exhibits obvious discontinuity because of its discrete characteristics of picking up $f$CO$_2$ values from the labelled SOM. For year 2005, the mean $f$CO$_2$ difference is -0.05±12.73 μatm for FNN-SVM and -0.6±18.80 for SOM-SVM. The uncertainty is the standard deviation of the mean difference between predicted and observed values. The statistics indicates that FNN agrees better with SVM than SOM does.

Although the differences among models might be on the order of 10 to 20 μatm, the effect on the global ocean CO$_2$ flux estimate is small (Fig.4). The fluxes are calculated using the wind speed from ECMWF's interim product (Dee et al., 2011). Our estimate for the oceanic uptake is on the higher end among those in Wanninkhof et al. (2013) and Le Quéré et al. (2015). For example, Wanninkhof et al. (2013) reported that the median sea–air anthropogenic CO$_2$ fluxes centered on year 2000 ranged from 1.9 to 2.5 PgC yr$^{-1}$ among the seven models. In comparison, our estimates by the three models are about 2.4 PgC yr$^{-1}$. The mean difference of CO$_2$ flux is 0.02 PgC yr$^{-1}$ between the FNN and the SVM (FNN-SVM) and 0.06 PgC yr$^{-1}$ between the SOM and the SVM (SOM-SVM). They are small in comparison with those differences among the models in Wanninkhof et al. (2013) and Le Quéré et al. (2015). Note that the flux estimate is highly dependent on wind products as shown by Wanninkhof et al. (2013) and Zeng et al. (2014).

On the spatial scale of tens of degrees, the three models show good mutual agreement for modelled $f$CO$_2$ distributions among them. However, each model shows distinguished fine structures, which are determined by the biogeochemical processes in the ocean, by model parameters obtained from training, and by the characteristics of the models. We believe that the modelled monthly $f$CO$_2$ distributions are true to the degree given by the model validations.

## 7 Summary

The main features of the three machine models are listed in Table 1. The SVM is recommended when the computer has enough memory to store the matrix in Eq.(A18), which is proportional to the square of the number of training data. The SVM performs the best, but the training time could become very long when the number of training data is too large to be handled by a computer without using virtual memory. For any given dataset, using the SVM requires a prior step to find the optimal value for the parameter $\sigma$ in Eq.(A10) and the parameter $\gamma$ in Eq.(A11).

The FNN model does not perform as well as the SVM, but the number of training data does not affect its training as much as the SVM's. The training time can become long when a large number of hidden neurons are used and many iterations are needed to achieve convergence. It takes longer time to train the FNN than the SVM for a small number of data points. However, the FNN is simpler to use as it requires no prior step. However, it may have the risk of over-extrapolation.

The SOM is recommended only when the other two models have over-fitting or over-interpolation problems. The SOM performs the worst and is not as straightforward as the others as its result depends too much on data scaling and the number of neurons. An advantage of the SOM is that once trained, re-labelling the SOM with new $CO_2$ measurements and making a new prediction is fast. Although the SOM does not have the over-extrapolation problem of the FNN, it may produce nonsense predictions due to its strong dependence on data scaling.

In areas where there was no measurement on a large scale, predictions made by the models must be treated conservatively, as SVM and FNN may produce extrapolated results and SOM may extract $CO_2$ from unexpected provinces. Figure 3 shows that the modelled $CO_2$ east of the African coast near the equator in July 2005 (Figure 3) appeared much higher than the nearby measurements, which were made in July 1995 and adjusted to 2005 using the global rate of 1.5 $\mu$atm yr$^{-1}$. However, considering the large variations of the rate from region to region (Takahashi et al., 2014) and of the repeated measurements discussed in section 5, the measurements were not sufficient to support rejecting the modelled $CO_2$. Similar $CO_2$ hotspots occurred in the Southern Ocean west of South America in February 2005, around the latitudinal zone of 50°S. The modelled $CO_2$ distributions by Takahashi et al. (2014) also showed $CO_2$ hotspots around the latitudinal zone of 30°S in the same month and region. Their model used a completely different interpolation scheme based on a diffusion–advection transport model for surface waters. In principle, these hotspot $CO_2$ were produced by our models using measurements somewhere else where the biogeochemical properties were similar to those in the hotspot areas. As the SOM does not make extrapolation, the SVM has low possibility of over-extrapolation, and the hotspots appeared in all models, the risk of accepting them would not be high.

**Appendix**

**A.1 Self-Organization Map**

A self-organizing map (SOM) is a type of artificial neural network that is trained using unsupervised learning (Kohonen, 1984). The SOM in our application comprises grid points on a two dimensional plane. Each grid point, also called neuron
cell, has the same number of parameters as the input variables, which include LAT, SST, SSS, CHL, MLD, and dSST in our case. Training the SOM is to use samples of input variables to adjust the parameters to make neighbourhood neuron cells having similar parameter values that reflect certain biogeochemical features of the surface ocean.

We used the batch learning algorithm (Abe et al., 2002) to train the SOM as the result does not depend on the sequential
order of training samples. The parameters were initialized randomly in the range (-1,1). In each iterative training loop, each training sample is associated with a neuron cell to which the distance defined as follow is smaller than to other neuron cells:

$$d = |\mathbf{f}(\mathbf{p} - \mathbf{x})|, \qquad (A1)$$

where $\mathbf{p}$ denotes the vector of neuron cell parameters, $\mathbf{x}$ the vector of input variables, and $\mathbf{f}$ the scale matrix that we introduced to change the influence of certain variables on the distance. The components of $\mathbf{f}$ are all zero except for those on
the diagonal, which are set to 1 by default. In our application, the data for each input variable were scaled to be unitless by its mean and standard deviation to eliminate the effect of units on the distance.

The associated neuron cell is called the best matching cell (BMC). After the BMC for all training samples are found, the parameters are updated by

$$p_i = \frac{\sum_k h_{ik} \mathbf{x}_k}{\sum_k h_{ik}}, \qquad (A2)$$

where $i$ and $k$ denote indexes of neuron cells and training samples, respectively. The neighbourhood function that determines the weight factor $h$ is defined as

$$h_{ik} = \exp(-\frac{|\mathbf{r}_{ik}|}{q}), \qquad (A3)$$

where $|\mathbf{r}_{ik}|$ denotes the geographic distance between the $i$th neuron cell and the BMC of the $k$th training sample and $q$ is a
factor that decreases linearly with iteration loop. In other words, the procedure adjusts the parameters of neuron cells toward those training samples whose BMC are close to them and the amount of adjustment decreases exponentially with the geographic distance between neuron cells and linearly with the training loop.

The trained SOM needs to be labelled by $f$CO$_2$ for making prediction. The values of $f$CO$_2$ measurements are assigned to their
BMC. Predicting $f$CO$_2$ for a set of input variables is realized by finding the BMC labelled with $f$CO2 and extract its mean $f$CO$_2$ value.

**A.2 Feedforward Neural Network**

A feedforward neural network (FNN) is an artificial neural network that is trained using supervised learning. Our FNN comprises three layers (Zeng et al., 2014): An input layer, a hidden layer, and an output layer. The number of neurons in the input layers is determined by the number of input variables, i.e., LAT, SST, SSS, CHL, MLD, and dSST in our case. The output layer has only one neuron for $f$CO$_2$. Each neuron in the hidden layer uses the following kernel function to transform all input variables:

$$y_h = \frac{1}{1+\exp(-(b+\mathbf{w}^\mathsf{T}\mathbf{x}))}, \qquad (A4)$$

where $\mathbf{w}$ denotes the vector of weight parameters and $b$ the offset parameter. The $y_h$ of all hidden neurons become the inputs of the output neuron, which uses the same kernel function to transform $y_h$ to produce $f$CO$_2$.

The training updates the offset and weight parameters, which are initialized randomly in the range (-1,1), by minimizing the cost function

$$f(\mathbf{w}') = \frac{1}{2}\mathbf{e}^T\mathbf{e} = \frac{1}{2}|\mathbf{y}_m - \mathbf{y}_o|^2. \qquad (A5)$$

where $\mathbf{w}'$ is the extended vector that include $b$ and $\mathbf{w}$; $\mathbf{y}_m$ and $\mathbf{y}_o$ stand for the vectors of modelled and observed $f$CO$_2$, respectively. In the gradient descent training algorithm, updating $\mathbf{w}'$ at the training iteration $t$ can be expressed as

$$\mathbf{w}'(\mathrm{t}) = \mathbf{w}'(t-1) - \alpha\mathbf{g} \qquad (A6)$$

where $\alpha$ is the learning rate (a positive number smaller than 1), and $\mathbf{g}$ the first-order derivative of the cost function:

$$\mathbf{g} = \nabla f(\mathbf{w}') = \mathbf{J}^\mathsf{T}\mathbf{e}, \qquad (A7)$$

where $\mathbf{J}$ is the Jacobian matrix whose components are derivatives of $\mathbf{e}$ with respect to $\mathbf{w'}$ using back propagation method. We used the efficient Levenberg-Marquardt algorithm (Wilamowski et al., 2010), which derives the gradient as

$$\mathbf{g} = (\mathbf{J}^T\mathbf{J} + \mu\mathbf{I})^{-1}\mathbf{J}^T\,\mathbf{e}, \qquad (A8)$$

where $\mu$ is a constant.

**A.3 Support Vector Machine**

A support vector machine (SVM) is a supervised learning model that was conceptualized in the 1960s for classification problems and later extended to regression analysis (Basak et al., 2007). We used the so called least-square support vector machine for regression (Pelckmans et al., 2002) which, similar to FNN, seeks to minimize the error between model outputs and measurements. The SVM models the dependence of $f$CO$_2$ on LAT, SST, SSS, CHL, MLD, and dSST as

$$y = \mathbf{c}^T\boldsymbol{\varphi}(\mathbf{x}) + b \qquad (A9)$$

where $\mathbf{x}$ stands for a set of measurements of the input variables, $\mathbf{c}$ the vector of coefficients, $b$ the offset parameter, and $\varphi$ the kernel function. In this investigation, we used the radial basis kernel function, i.e.,

$$\boldsymbol{\varphi}(\mathbf{x}_i)^T \boldsymbol{\varphi}(\mathbf{x}_j) = \exp\left(-\frac{|\mathbf{x}_i - \mathbf{x}_j|^2}{2\sigma^2}\right), \qquad (A10)$$

where $\sigma$ is a parameter whose optimal value depends on the data used for training. The subscription of $\mathbf{x}$ indicates a sample of input variables.

Given a set of training samples $\{\mathbf{x}_k, y_k\}_{k=1}^N$, the goal of training SVM is to minimizes the cost function

$$F(\mathbf{c}) = \frac{1}{2}(\mathbf{c}^T \mathbf{c} + \gamma \mathbf{e}^T \mathbf{e}) \qquad (A11)$$

where

$$e_k = y_k - \mathbf{c}^T \boldsymbol{\varphi}(\mathbf{x}_k) - b \qquad (A12)$$

and $\gamma$ is a constant whose optimal value depends on the data used for training. The Lagrangian solution for the optimization 
problem of Eq.(A11) is given by

$$L(c, e, b, \alpha) = \frac{1}{2}(\mathbf{c}^T \mathbf{c} + \gamma|e|) - \sum_k^N \alpha_k\{\mathbf{c}^T \boldsymbol{\varphi}(\mathbf{x}_k) + b + e_k - y_k\}, \qquad (A13)$$

where $\alpha_k$ is a Lagrangian multiplier. The optimal conditions of Eq.(A13) are:

$$\frac{\partial L}{\partial c_k} = 0 \;\; \rightarrow \;\; c_k = \alpha_k \varphi(\mathbf{x}_k), \qquad (A14)$$

$$\frac{\partial L}{\partial b} = 0 \;\; \rightarrow \;\; \sum_k^N \alpha_k = 0, \qquad (A15)$$

$$\frac{\partial L}{\partial e_k} = 0 \;\; \rightarrow \;\; \alpha_k = \gamma e_k, \qquad (A16)$$

$$\frac{\partial L}{\partial \alpha_k} = 0, \;\; \rightarrow \;\; c_k \varphi(\mathbf{x}_k) + b + e_k - y_k = 0, \quad (A17)$$

After eliminating $c$ and $e$ from the above conditions, the following equation is obtained:

$$\begin{bmatrix} 0 & \mathbf{u}^T \\ \mathbf{u} & \boldsymbol{\Omega} + \gamma^{-1}\mathbf{I} \end{bmatrix} \begin{bmatrix} b \\ \boldsymbol{\alpha} \end{bmatrix} = \begin{bmatrix} 0 \\ \mathbf{y} \end{bmatrix}, \qquad (A18)$$

where $\mathbf{u}$ is a vector with all components being 1, and the components of $\boldsymbol{\Omega}$ are

$$\Omega_{ij} = \boldsymbol{\varphi}(\mathbf{x}_i)^T \boldsymbol{\varphi}(\mathbf{x}_j). \qquad (A19)$$

Once Eq.(A18) is solved, making a prediction is done by

$$y(x) = \sum_k^N \alpha_k \boldsymbol{\varphi}(\mathbf{x}_k)^T \boldsymbol{\varphi}(\mathbf{x}) + b \qquad (A20)$$


**Acknowledgements**

The Surface Ocean CO2 Atlas (SOCAT) is an international effort, endorsed by the International Ocean Carbon Coordination Project (IOCCP), the Surface Ocean Lower Atmosphere Study (SOLAS) and the Integrated Marine Biogeochemistry and Ecosystem Research program (IMBER), to deliver a uniformly quality-controlled surface ocean CO2 database. The many researchers and funding agencies responsible for the collection of data and quality control are thanked for their contributions to SOCAT.

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

Table 1 Feature comparison of the three machine learning models.

| Feature | SVM | FNN | SOM |
|---|---|---|---|
| Input space projection | Projects the input variable space to a high dimensional space that is proportional to the number training samples. | Projects the input space to a high dimensional space that is proportional to the number of hidden neurons and input variables. | Projects the input space to a feature space whose size is determined by the number of neurons. |
| Prediction by | Continuous interpolation. | Continuous interpolation. | Picking up labelling samples that have the closest feature to the input. |
| Problems | May over-fit and over-interpolate. | May over-fit and over-interpolate. | Discrete interpolation leads to spatial discontinuity. |
| Data scaling | Helps solving the linear equation, but has no effect on the result. | Helps the convergence of training, but has insignificant effect on the result. | Significant effect on the result. |
| Results affected by | The parameter values for regularization and kernel function. | The number of hidden neurons. | The number of neurons and data scaling. |

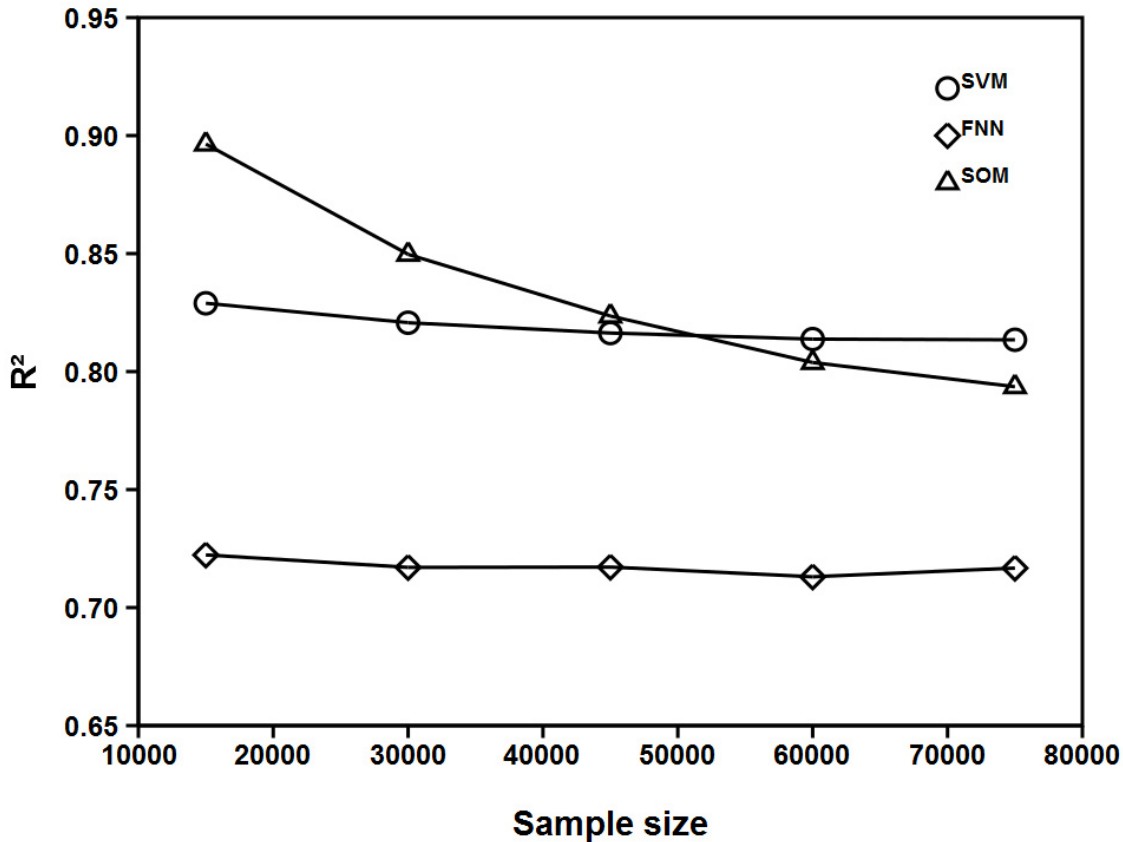

Figure 1: Correlation coefficient between modelled and observed $f$CO$_2$ (uatm). The sample size is the number of data points randomly selected to train FFN and SVM and to label SOM.

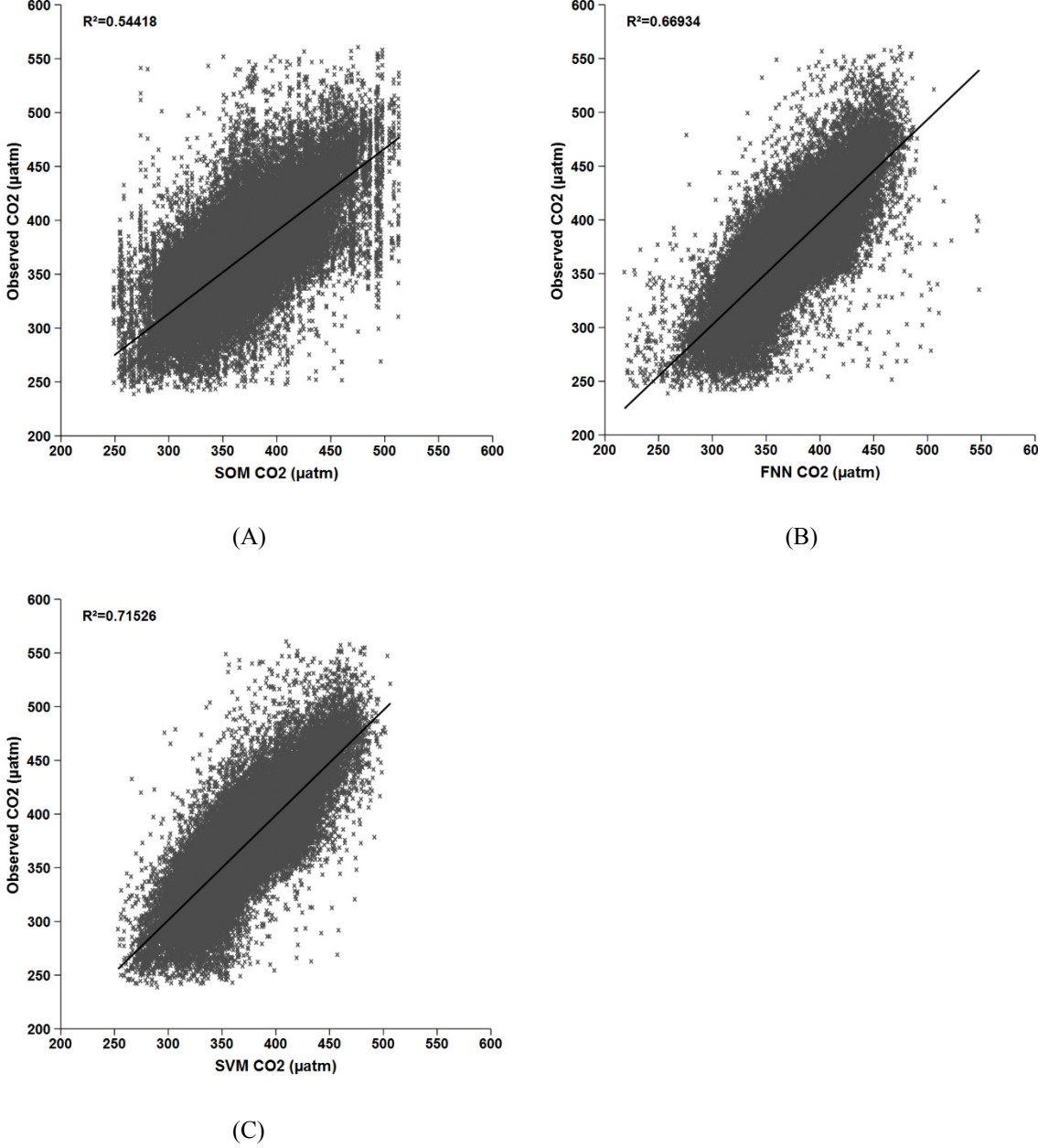

(A)

(B)

(C)

5   Figure 2: Predicted vs observed $f\text{CO}_2$ (µatm). Ten percent of data points was selected randomly to train FNN and SVM and to label SOM, and the rest was used for validation.

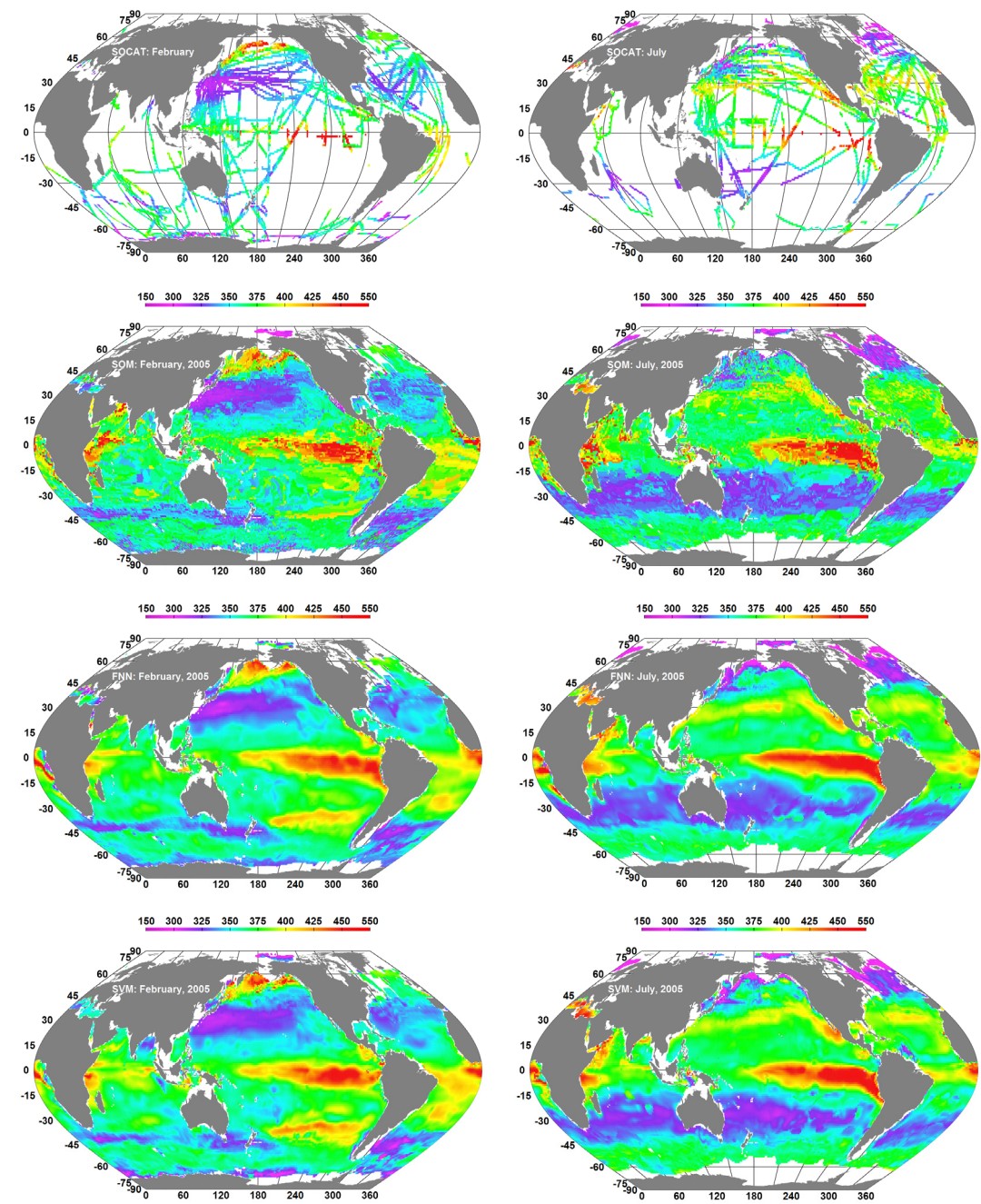

5    Figure 3: Distributions of modelled and observed $f$CO$_2$. The composite map for observations includes $f$CO$_2$ in 1990-2014. Half of randomly selected data points were used to train FNN and SVM and to label SOM to make prediction. The left panels show February and the right panels show July.

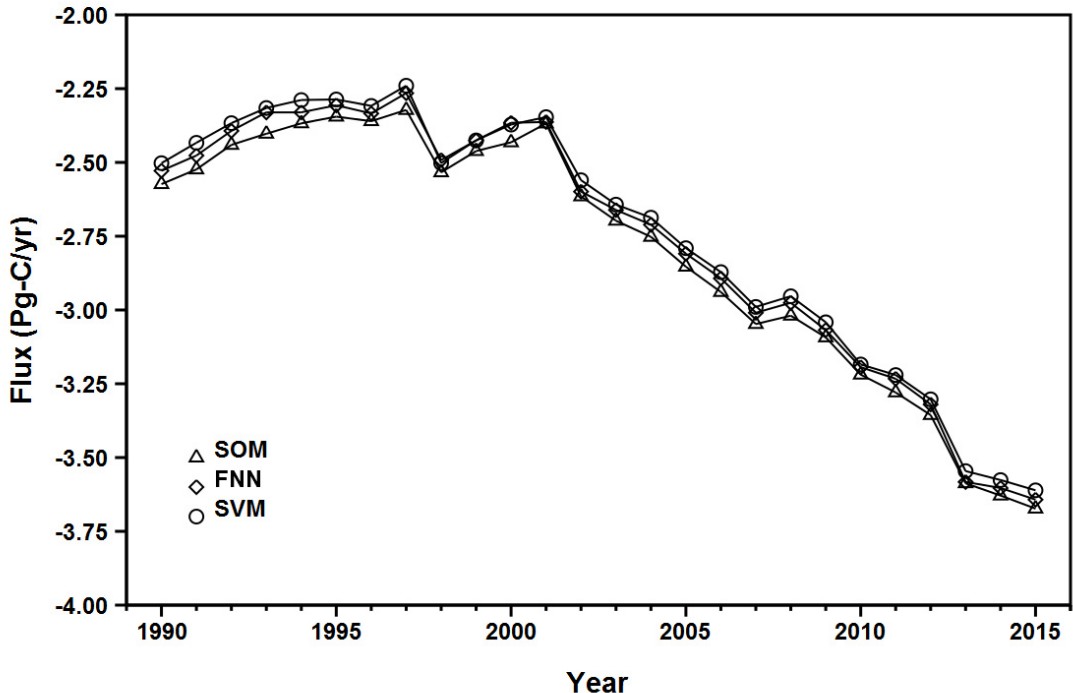

Figure 4: Modelled global $CO_2$ fluxes. A negative value indicates oceanic uptake.