# Peer review of "Technical note: Evaluation of three machine learning models for surface ocean CO2 mapping"

_Ocean Science, 2016_

## Referee Comment (RC1) · Anonymous Referee #1 · 29 Nov 2016

This Technical note compares the results of three machine learning models for sea surface CO2 mapping. Two of those, self-organizing-maps (SOM) and feedforward neural networks (FNN), have already been used and compared (in the Surface Ocean CO2 Mapping inter comparison initiative, SOCOM) and a new one, the support vector machine (SVM), is introduced in this paper. The SVM performs best but requires big computer memory.

This is valuable work as with ever increasing computer power SVM will become available to more users. I have one concern: the resulting model distributions show features that cannot be explained by the CO2 field data. For example there is a CO2 hotspot east of the African coast near the equator where no observations (February) or low CO2 observations (July) are shown in the top panel. In July there is an unexplained hotspot in the Southern Ocean west of South America where there are no observations. I presume these features are produced by the correlation of sea surface CO2 with proxy variables such as SST, SSS CHL and MLD? Are these hotspots known / expected from previous publications? The authors should discuss this further in the discussion of Figure 3.

My final question is: is the dataset produced by SVM available for download somewhere or can it be retrieved from the authors? Could this be added as a supplement possibly?

Detailed comments

Page 1, line 14: include (Goddijn-Murphy et al, 2015).

Page 2, line 13: please explain "circular property" and why it can therefore not be used.

Page 2, line 14: sine and cosine transformed components of LON and MON? How of MON?

Page 2, line 14: "The approach..." is meaning "Our approach..." or "Zeng et al.'s approach..."?

Page 3, line 4: which two CHL products, calculated from OC3 and OCI algorithms?

Page 3, line 8, refer to Table 1 here

Page 3, line 11: 10% of the measurements randomly chosen?

Page 3, line 12: "dependent of" should be "dependent on"

Page 3, line 17: insert ", $\nu$," in "all variables, $\nu$,"; explain all variables (SST, SSS, CHL, MLD, dSST?).

Page 4, line 2: give references for preliminary studies

Page 4, line 13: replace "to model" with "and modelled"

Page 4, line 18: modeled and observed CO2 of "all / selected/ non-selected" data

points?

Page 5, line 6: random 10%?

Page 5, line 8: differences are expressed as mean difference $\pm$ standard deviation?

Page 5, line 8: replace "respectively" with "for SOM"

Page 5, line 9: give range of measurement uncertainties, how small is small?

Page 5, line 15-17, Fig. 3: The panels for both February and July show features in all three model distributions that are not seen in the field $CO_2$. For example there is a hotspot on the eastern African coast in the western Indian Ocean that is not seen in the observations (top panel). Likewise in July there is an unexplained hotspot west of South America in the Southern Ocean. So, "the models captured the major features of spatial distribution of observed $CO_2$" plus quite a bit more. Can the authors discuss this further in page 5, line 30 - page 6, line 2?

Page 8, line 8: "prediction" should be "predictions ".

Acknowledgements

Include, as suggested on SOCAT's website:

"The Surface Ocean CO2 Atlas (SOCAT) is an international effort, endorsed by the International Ocean Carbon Coordination Project (IOCCP), the Surface Ocean Lower Atmosphere Study (SOLAS) and the Integrated Marine Biogeochemistry and Ecosystem Research program (IMBER), to deliver a uniformly quality-controlled surface ocean CO2 database. The many researchers and funding agencies responsible for the collection of data and quality control are thanked for their contributions to SOCAT."

Table 1: Add a first column 'Feature', e.g., 1-input space mapping, 2-prediction by, 3-problems, 4-data scaling, 5-results affected by. Then revise the SVM, FNN, SOM columns accordingly.

Table 1, line 9: 'closet' should be 'closest'

Figure 3: The labels in white font are too small to read.

References Goddijn-Murphy, L. M., Woolf, D. K., Land, P. E., Shutler, J. D., Donlon, C. (2015) The OceanFlux Greenhouse Gases methodology for deriving a sea surface climatology of CO2 fugacity in support of air–sea gas flux studies. Ocean Science 11: 519-541. doi:10.5194/os-11-519-2015.

---

## Referee Comment (RC2) · Anonymous Referee #2 · 23 Jan 2017

This "technical note" discusses the formation of global maps of surface ocean CO2 from limited measurements using inferred dependence on (latitude, surface temperature SST, salinity, chlorophyll concentration, mixed-layer depth, difference between monthly- and annual-mean SST). The dependence is inferred by three methods: self-organisation map (SOM), feedforward neural network (FNN) and a new method (support vector machine; SVM). The results of these three methods, "trained" on a fraction of the data, are compared with the remaining data. The correlations are not particularly good for any (best at R2 = 0.715 for SVM) considering there are 6 independent variables aiding the fit. However, the results of all three methods for global air-sea CO2 flux are very close and the CO2 maps are visually similar. This similarity extends to a band of high CO2 concentration in February 2005 extending west from Chile where there are apparently no CO2 measurements. This extrapolation from CO2 observations is

presumably via a similar feature in (at least) one of the 6 independent variables.

There should be more discussion: (i) of the quality of the fit to observed data, especially in relation to the estimates of air-sea flux and the danger that the methods agree with each other more than with reality; (ii) of the extrapolation feature west of Chile (in particular – perhaps also a careful examination for whether there are others) and whether it can be believed in terms of the values of the independent variables – is this set of six values closely approximated somewhere else where there are CO2 measurements constraining the CO2 estimate?

Although the organisation and English are generally good, I think some sections and especially the Appendix are unclear/obscure, mainly due to inconsistent or missing explanations, definitions or notation. Most of the following detailed comments are about this aspect.

(Section 2)

Page 2 lines 12 and 18. "dSST denotes the difference between the monthly and annual means of SST" implies 12 discrete values of dSST; how does this "improve expressing the seasonal variable continuously"?

(Section 4)

Line 13. I think you mean ". . to the range (0, 1) for the SVM . . ."

Line 21 (i.e. line after (5)). Why between 0.1 and 0.9 not between 0 and 1? "better" compared with what? Why should scaling the output help?

Lines 1-2. "We used Eq. (4) to scale . . SOM". There is no mention of this in Appendix A.1, indeed after (A1) it is stated that the diagonal factors of the scale matrix f are equal to 1.

Lines 2-3. "Based on preliminary studies, we applied a factor of 2 to . . SST and CHL . .". What preliminary studies? Is this subjective, i.e. why should SST and CHL be emphasised?

Line 7. "prediction for an input" needs explaining. Inputs are supposed to be known, not "predicted".

Lines 8, 9. "map size". In normal language the map size is the earth's surface area. Do you mean resolution, equivalent to the number of $CO_2$ output locations? Please explain / use correct word.

(Section 5)

Line 8. "respectively" should be "for SOM"

Lines 11, 15. Please explain "normalized"/"normalization"

Appendix, page 6. To have value, this needs to be understood in its own terms; the reader should not have to refer to cited references to understand the words used and the overall meaning. Too many words are not defined or explained. Also, it is too abstract. This is a manuscript about "output" $CO_2$, depending on "inputs" LAT, SST, SSS, CHL, MLD, dSST. Presumably this applies to A.1, A.2 and A.3 – say so and do not use vague terms like "feature space" – at present the reader has to guess what you mean.

(A.1 . .)

Line 23. What is "feature space" in oceanographic terms?

Lines 23-24. "usually represented by grid points in two dimensional space". Never mind about "usually"; describe in terms of the problem here.

Line 24. "weight vector w". This name is confusing. On page 7 lines 7-8 weights (weight factors) h are defined by (A3). "w" is the result of applying the weights "h" to combine values of "v" at various locations [presumably to represent "v" at grid locations rather than original locations, but this is not clear to the reader. If this the case, then "w" is "gridded v" or "interpolated v"]. See also the comment on page 7 line 21.

Line 25. Not "a data vector" which might refer to any vector at all, but "an input data vector" (I guess).

Line 30. "best matching cell (BMC)" needs explaining.

Line 30. "minimizing the distance". What is varied to do this?

Line 4. "matched". Either this is the wrong word or it needs explaining.

(A.2 . .)

Line 17. "vector x of input data". In A.1 the input data were "v". Use consistent names for variables.

Lines 20-22. You have input data, hidden neurons and output. There should be distinct variable names for each of these, e.g. v, x, y respectively. Here you have y for the hidden neurons and for the output, which is confusing.

Line 21. "w is the weight vector". Indeed this seems correct for its use in (A4) but that is very different from its use in (A1). Use different terms for different quantities (c.f. comment on page 6 line 24).

Line 22. "The training updates the offset and weight parameters". What are the starting values before updating? Do you mean "weight vector" as in line 21?

Line 23. What is "e" or is it defined by (A5)? Please make this clear.

Line 24. "modelled . . y" is unclear (especially because you use "y" for hidden neurons and output). Why are two "y" in this line in bold type but not the third or "y" in (A4)?

Line 24. "w includes both . ." This seems to be defining a vector with more components; it should have a new name.

Line 28. "$\alpha$ is the learning rate". How is its value decided?

Line 30. "derivatives of e by w". Do you mean "derivatives of e with respect to w".

(A.3 . .)

Lines 6-10. "The SVM . . SVM parameters." Is this relevant?

Line 14. "independent variables", "high dimensional space", "target variable". Please define these in terms of the oceanographic problem in question.

Line 16. "minimizes" – what is varied to do this?

Lines 18-19. "subjecting to the constraint". (A11) looks like a definition of "e" and is not a constraint unless "e" is defined in some other way which needs to be stated.

Line 27. Can there be an explicit expression for $\varphi$? Where has "b" in (A9) gone to?

Table 1. SOM column half way down. "closest" not "closet"!

Figure 3 caption. Please explain "normalized to 2005".

---

## Referee Comment (RC3) · Anonymous Referee #3 · 26 Feb 2017

The basis of the this technical note is a more detailed examination of the application of various configurations of the neural network approach to create monthly maps of sea surface pCO2 from temporally and spatially sparse observations. These look to build predictive relationships from high resolution satellite sea surface temperature (SST), chlorophyll a, salinity, mixed layer depth and datasets and SST anomaly. Here, the authors focus on two previously explored methodologies (involving self-organising maps - SOM, and feed-forward neural networks - FNN) in addition to a third novel approach (support vector machine - SVM). This is a timely study, as neural network-based approaches have found increasing application and influence within the community. A thorough investigation of the advantages and weaknesses of individual configurations is an important next step.

Unfortunately, this study is not able to achieve this. It is quite a threadbare investigation,

and spends more time comparing outputs from each configuration against each other, rather than the more important comparison against real observations, previous studies, and alternative independent methods. More attention should be made describing the exact application of the method to the data, the assumptions made and their impact, an assessment of the true uncertainty on pCO2 mapping, and how and why the estimates diverge from observations / other methods. For instance, has a riverine fCO2 flux been accounted for in the flux estimates presented here, as this needs to be done in order to be compare like for like

General points: More detail of the exact data application steps are required: Did the application of the methods follow the biogeochemical province-by-province approach of SOCOM, or was all global data combined together? A comment regarding the use of a single trend normalization rate would be welcome. It is known that this is not globally uniform (e.g. Takahashi et al., 2014) and so it would be good to understand the impact of this choice. Why are the correlations so much poorer than that achieved by the application of the SOM-FFN approach of Landschutzer et al, 2014)? Within the model validation section, was the random selection of 50% data carried out only once or multiple times? What is the effect of this random selection compared to say, using data clustered around 2005, or only data from regions where pCO2 varies the most, or only using the most recent data? I would imagine this would be useful information for other researchers looking to apply the methods themselves, whether to map sea surface pCO2 or indeed other biogeochemical parameters. As mentioned above, the study would benefit with comparison with independent dataset e.g. time series at BATS / HOTS. There is very little coverage on uncertainties. More detail on how these are calculated, especially for regions where there are no observational data with which to compare (e.g. South Pacific / Southern Ocean) would be very welcome. This could useful be useful in explaining the anomolous flux feature currently prevalent in Figure 3 in the South Pacific, which is not mentioned in the text and does not appear to be supported by observations or previous studies (e.g. the Takahashi climatology). There are substantial

Structure: I feel it would be better to have the methodological description section currently situated within the appendix to be within the main body of the text. To a non-user of neural networks, it seems disingenuous to direct readers to the end of the manuscript in order to understand the details underpinning the outputs.

Figures: - Figure 2 - unity line is not easily seen. Possibly changing the color of data points to gray could remedy this? - Figure 3 - needs larger labelling as to what they are showing. A column title would be useful, and a more color-blind friendly colorscale.

Specific points: p5 l7 - what do the uncertainties represent? Are these the standard error of the fit, standard deviation of the mean difference between predicted and observed values? How do these compare to other non neural network methods applied during SOCOM? p5 l9 - what are the measurement uncertainties? p5 l10 - what is this uncertainty from temperature? p5 l11 - what is the average standard deviation of repeat measurements (should also reference) p5 l13 - why is only july looked at, what is the uncertainty for the full year? How much of this is due to the normalization method? p5 l25 - there seems some agreement with other studies for 2000 but substantial disagreement with other estimates (Wanninkhof et al., 2013, Rodenbeck et al., 2015) for 2010. This is surprising given that this is when there are most observational data and so it could be assumed that this era would be best modelled. Equally it is rather worrying that the same models as used in the SOCOM study are showing substantially higher estimates for the air-sea CO2 flux for the same input dataset. Is this related to the choice of wind field or how the mapped pCO2 fields are built? How do the mapped pCO2 fields compare with other methods? Some comment on this discrepancy would be greatly appreciated. In particular, comment on how fluxes for years other then 2000 are calculated would be useful as this is not currently explained. Is the systematic trend of 1.5uatm/year simply reintroduced. p5 l27 - the within-model differences are smaller, but this would be expected as they are essentially iterations of a similar technique. More disconcerting is the substantial offset of this group of models with other independent approaches. As mentioned above, more comment/discussion

on this aspect would be useful

---

## Author Comment (AC1) · 2 Mar 2017

We thank referee#1 for the thoughtful and constructive feedback on the paper. We have addressed major concerns in the revised manuscript and documented our responses to the referee's comments point-by-point as follows.

Q1. This Technical note compares the results of three machine learning models for sea surface CO2 mapping. Two of those, self-organizing-maps (SOM) and feedforward neural networks (FNN), have already been used and compared (in the Surface Ocean CO2 Mapping inter comparison initiative, SOCOM) and a new one, the support vector machine (SVM), is introduced in this paper. The SVM performs best but requires big computer memory. This is valuable work as with ever increasing computer power SVM will become available to more users. I have one concern: the resulting model distributions show features that cannot be explained by the CO2 field data. For example there is a CO2 hotspot east of the African coast near the equator where no observations (February) or low CO2 observations (July) are shown in the top panel. In July there is an unexplained hotspot in the Southern Ocean west of South America where there are no observations. I presume these features are produced by the correlation of sea surface CO2 with proxy variables such as SST, SSS CHL and MLD? Are these hotspots known / expected from previous publications? The authors should discuss this further in the discussion of Figure 3.

Reply: Referee #1 pointed out the CO2 hotspot east of the African coast near the equator and suggested a further discussion. The hotspot CO2 seems quite high comparing to the nearby measurement made by the one cruise in July 1995. It is very difficult to judge whether this is an issue because the observed CO2 shown in the figure is trend removed with a universal rate, which could be another source of uncertainty. Takahashi et al. (2009) shown that the trend could be quite different in different areas. However, it difficult to use multiple rates for global mapping. As the issue cannot be cleared without thorough numerical comparisons to other models's output of the same spatial and time resoutions and extend this technical note to a full research paper, it is beyond the scope of this manuscript, which is aimed at comparing the three machine learning models.

Q2. My final question is: is the dataset produced by SVM available for download somewhere or can it be retrieved from the authors? Could this be added as a supplement possibly?

Reply: We uploaded the data sets of all models as supplement.

Q3. Page 1, line 14: include (Goddijn-Murphy et al, 2015).

Reply: We included the reference.

Q4. Page 2, line 13: please explain "circular property" and why it can therefore not be

used.

Reply: In the manuscript we added "For instance, longitude -180 degree is geographically connected to longitude 180 degree, but numerically they appear to be two extreme longitude values to the models."

Q5. Page 2, line 14: sine and cosine transformed components of LON and MON? How of MON?

Reply: The transforms are $\cos(MON*2*pi/12)$, $\sin(MON*2*pi/12)$, $\cos(Lon*2*pi/360)$ and $\sin(LON*2*pi/360)$. See Zeng et al. (2015). We didn't give the transform here because MON and LON were not used.

Q6. Page 2, line 14: "The approach" is meaning "Our approach" or "Zeng et al.'s approach"?

Reply: We revised "The approach ..." to "Their approach ..." to explicitly mean Zeng et al.'s approach.

Q7. Page 3, line 4: which two CHL products, calculated from OC3 and OCI algorithms?

Reply: The footnote indicates that the products used the OCI algorithm

Q8. Page 3, line 8, refer to Table 1 here

Reply: We revised "The Appendix summarizes..." to "The Appendix and Table 1 summarize...".

Q9. Page 3, line 11: 10% of the measurements randomly chosen?

Reply: Yes, they are randomly chosen. We revised "we used 10% of..." to "we randomly chose 10% of..."

Q10. Page 3, line 12: "dependent of" should be "dependent on".

Reply: We corrected the mistake.
Q11. Page 3, line 17: insert "" in "all variables "; explain all variables (SST, SSS, CHL, MLD, dSST?).

Reply: We revised the sentence to "we scaled all input variables LAT, SST, SSS, CHL, MLD, and dSST by their minimum and maximum to confine them in the range (0, 1)".

Q12. Page 4, line 2: give references for preliminary studies.

Reply: We revised "Based on preliminary studies" to "Based on our preliminary correlation analysis".

Q13. Page4, line 13: replace "to model" with "and modelled".

Reply We revised the expression accordingly.

Q14. Page 4, line 18: modeled and observed CO2 of "all / selected/ non-selected" data points?

Reply: We added "of the selected data points" to the end of the sentence.

Q15. Page 5, line 6: random 10%?

Reply: Yes, they were randomly selected. We revised "with 10% of the data" to "with 10% of randomly selected data points".

Q16. Page 5, line 8: differences are expressed as mean difference standard deviation?

Reply: Yes.

Q17. Page 5, line 8: replace "respectively" with "for SOM".

Reply: We corrected the mistake.

Q18. Page 5, line 9: give range of measurement uncertainties, how small is small?

Reply: In the revision we add information for the standard deviation of gridded data for the discussion.

Q19. Page 5, line 15-17, Fig. 3: The panels for both February and July show features in all three model distributions that are not seen in the field CO2. For example there is a hotspot on the eastern African coast in the western Indian Ocean that is not seen in the observations (top panel). Likewise in July there is an unexplained hotspot west of South America in the Southern Ocean. So, "the models captured the major features of spatial distribution of observed CO2" plus quite a bit more. Can the authors discuss this further in page 5, line 30 - page 6, line 2?

Reply: See the reply to question 1.

Q20. Page 8, line 8: "prediction" should be "predictions ".

Reply: We corrected the mistake.

Q21. Acknowledgements. Include, as suggested on SOCAT's website: "The Surface Ocean CO2 Atlas (SOCAT) is an international effort, endorsed by the International Ocean Carbon Coordination Project (IOCCP), the Surface Ocean Lower Atmosphere Study (SOLAS) and the Integrated Marine Biogeochemistry and Ecosystem Research program (IMBER), to deliver a uniformly quality-controlled surface ocean CO2 database. The many researchers and funding agencies responsible for the collection of data and quality control are thanked for their contributions to SOCAT.

Reply: We revised the acknowledgements as suggested.

Q22. Table 1: Add a first column 'Feature', e.g., 1-input space mapping, 2-prediction by, 3-problems, 4-data scaling, 5-results affected by. Then revise the SVM, FNN, SOM columns accordingly.

Reply: We revised the table as suggested.

Q23. Table 1, line 9: 'closet' should be 'closest'.

Reply: We corrected the mistake.

Q24. Figure 3: The labels in white font are too small to read.

Reply: We enlarged the labels.

Please also note the supplement to this comment:
http://www.ocean-sci-discuss.net/os-2016-73/os-2016-73-AC1-supplement.zip

────────────────────────────────

---

## Author Comment (AC2) · 2 Mar 2017

We thank referee#2 for the thoughtful comments, especially on the appendix. We have revised the appendix substantially to address the reviser's constructive opinions. Here we documented our responses to the reviewer's comments point-by-point.

Q1. This "technical note" discusses the formation of global maps of surface ocean CO2 from limited measurements using inferred dependence on (latitude, surface temperature SST, salinity, chlorophyll concentration, mixed-layer depth, difference between monthly- and annual-mean SST). The dependence is inferred by three methods: selforganisation map (SOM), feedforward neural network (FNN) and a new method (support vector machine; SVM). The results of these three methods, "trained" on a fraction of the data, are compared with the remaining data. The correlations are not particularly good

for any (best at R2 = 0.715 for SVM) considering there are 6 independent variables aiding the fit. However, the results of all three methods for global air-sea CO2 flux are very close and the CO2 maps are visually similar. This similarity extends to a band of high CO2 concentration in February 2005 extending west from Chile where there are apparently no CO2 measurements. This extrapolation from CO2 observations is presumably via a similar feature in (at least) one of the 6 independent variables. There should be more discussion: (i) of the quality of the fit to observed data, especially in relation to the estimates of air-sea flux and the danger that the methods agree with each other more than with reality; (ii) of the extrapolation feature west of Chile (in particular – perhaps also a careful examination for whether there are others) and whether it can be believed in terms of the values of the independent variables – is this set of six values closely approximated somewhere else where there are CO2 measurements constraining the CO2 estimate?

Reply: First on the quality of fit. As the three model are unbiased, i,e, the mean difference between modeled CO2 and observation is statistically zero, we take the quality here means the correlation and the standard deviation between modeled CO2 and observation. Then quality is not only determined by the capability of the models, but also the variability of the data (we added this information in the revised manuscript). Second on the relation to the estimates of air-sea flux. The models may produce differently higher than observed CO2 in some areas and lower CO2 in others. The flux could be largely affect by this and by different wind. That all three models produced similar global fluxes indicates that the effect is small. Third on the danger that the methods agree with each other more than with reality. They are quite different issues. The answer to the quality of fit indicates the models cannot agree with reality than the variability of the reality. Whereas, the agreement between models is determined mainly by their similarity. For example, SVM is considered to be a one layer FNN in some articles; so FNN agrees better with SVM than with SOM. Fourth on the extrapolation. Yes, the extrapolation approximate unmeasured area with somewhere that has a similar biogeochemical property and CO2 measurement.

Q2. Although the organisation and English are generally good, I think some sections and especially the Appendix are unclear/obscure, mainly due to inconsistent or missing explanations, definitions or notation. Most of the following detailed comments are about this aspect.

Reply: We have revise the appendix substantially to address to issue.

Q3. Page 2, lines 12 and 18. "dSST denotes the difference between the monthly and annual means of SST" implies 12 discrete values of dSST; how does this "improve expressing the seasonal variable continuously"?

Reply: Let's consider three measurements taken on January 1, January 31, and February 1. Using month as the seasonal variable, the variable values of the first two measurements are 1 and the last is 2. However, the seasonally the last two are nearly the same. dSST reflects better the actual change of seawater property caused by season change.

Q4. Page 3, Line 13. I think you mean ". . to the range (0, 1) for the SVM . . ."

Reply: We revised the expression accordingly.

Q5. Page 3,. Line 21 (i.e. line after (5)). Why between 0.1 and 0.9 not between 0 and 1? "better" compared with what? Why should scaling the output help?

Reply: For fCO2 close to 0 and 1, and a small change in fCO2 requires very large adjustment of model parameters, which slows down the convergence of training. We added this in the revised manuscript.

Q6. Page 4, Lines 1-2. "We used Eq. (4) to scale . . SOM". There is no mention of this in Appendix A.1, indeed after (A1) it is stated that the diagonal factors of the scale matrix f are equal to 1.

Reply: In the appendix section for SOM, we added "In our application, the data for each input variable were scaled to be unitless by its mean and standard deviation".

Q7. Page 4, Lines 2-3. "Based on preliminary studies, we applied a factor of 2 to . . SST and CHL . .". What preliminary studies? Is this subjective, i.e. why should SST and CHL be emphasised?

Reply: SOM is indeed subjective. In our knowledge, other applications scaled the data with different subjective factors to change the impact of independent variables on the distance defined by Eq.(A1). In our application, we scaled the data non-subjectively and uses the scale factors to change the impact, which in our opinion is easier to understand. Because $CO_2$ shows a much higher correlation with SST and CHL than with others, we subjectively used a factor of 2 for them. There is no theoretical basis for this choice. We revised the manuscript to address the issue and revised "Based on preliminary studies" to "Based on our preliminary correlation analysis"

Q8. Page 4, Line 7. "prediction for an input" needs explaining. Inputs are supposed to be known, not "predicted".

Reply: We revised "Making prediction for an input" to "Making a $CO_2$ prediction for an input"

Q9. Page 4, Lines 8, 9. "map size". In normal language the map size is the earth's surface area. Do you mean resolution, equivalent to the number of $CO_2$ output locations? Please explain / use correct word.

Reply: We revised "the feature map size" to "the number of neuron cells"

Q10. Page 5, Line 8. "respectively" should be "for SOM"

Reply: We corrected the mistake.

Q11. Page 5, Lines 11, 15. Please explain "normalized"/"normalization"

Reply: We revised the manuscript and explained "normalized"/"normalization".

Q12. Page 6. To have value, this needs to be understood in its own terms; the reader should not have to refer to cited references to understand the words used and the

overall meaning. Too many words are not defined or explained. Also, it is too abstract. This is a manuscript about "output" CO2, depending on "inputs" LAT, SST, SSS, CHL, MLD, dSST. Presumably this applies to A.1, A.2 and A.3 – say so and do not use vague terms like "feature space" – at present the reader has to guess what you mean.

Reply: We have revised the appendix substantially according to address the concern.

Q13. (A.1 . .) Page 6 Line 23. What is "feature space" in oceanographic terms?

Reply: We removed the jargon.

Q14. Lines 23-24. "usually represented by grid points in two dimensional space". Never mind about "usually"; describe in terms of the problem here.

Reply: We revised the expression to be specific.

Q15. Line 24. "weight vector w". This name is confusing.

Reply: We changed the symbol and revised the descriptions.

Q16. page 7, lines 7-8 weights (weight factors) h are defined by (A3). "w" is the result of applying the weights "h" to combine values of "v" at various locations [presumably to represent "v" at grid locations rather than original locations, but this is not clear to the reader. If this the case, then "w" is "gridded v" or "interpolated v"]. See also the comment on page 7 line 21. Line 25. Not "a data vector" which might refer to any vector at all, but "an input data vector" (I guess).

Rely: We changed the symbol and revised the descriptions.

Q17. page 7, Line 30. "best matching cell (BMC)" needs explaining.

Reply: We revised the descriptions.

Q18. page 7, Line 30. "minimizing the distance". What is varied to do this? Reply: We revised the descriptions.

Q19. Page 7 Line 4. "matched". Either this is the wrong word or it needs explaining.
[Figure]

Reply: We revised the descriptions.

Q20. Line 17. "vector x of input data". In A.1 the input data were "v". Use consistent names for variables.

Reply: We revised the descriptions.

Q21. Lines 20-22. You have input data, hidden neurons and output. There should be distinct variable names for each of these, e.g. v, x, y respectively. Here you have y for the hidden neurons and for the output, which is confusing.

Reply: We changed the symbols and revised the descriptions.

Q22. Line 21. "w is the weight vector". Indeed this seems correct for its use in (A4) but that is very different from its use in (A1). Use different terms for different quantities (c.f. comment on page 6 line 24).

Reply: We changed the symbols and revised the descriptions.

Q23. Line 22. "The training updates the offset and weight parameters". What are the starting values before updating? Do you mean "weight vector" as in line 21?

Reply: We revised the descriptions. The parameters are initialized randomly between -1 and 1. We added this information in the revised manuscript.

Q24. Line 23. What is "e" or is it defined by (A5)? Please make this clear.

Reply: Yes. It is the "e" defined b (A5).

Q25. Line 24. "modelled . . y" is unclear (especially because you use "y" for hidden neurons and output). Why are two "y" in this line in bold type but not the third or "y" in (A4)?

Reply: We revised the description. Bold font indicate vector or matrix.

Q26. Line 24. "w includes both . ." This seems to be defining a vector with more components; it should have a new name.

Reply: We used a new name.

Q27. Line 28. " is the learning rate". How is its value decided?

Reply: The initial value is about 0.25. It is determined by try-and-error. A small value make training slow. A large value make a training diverge. We added the information in the revised manuscript.

Q28. Line 30. "derivatives of e by w". Do you mean "derivatives of e with respect to w".

Reply: We revised the description.

Q29. Page 8, Lines 6-10. "The SVM . . SVM parameters." Is this relevant?

Reply: We removed this part.

Q30. Line 14. "independent variables", "high dimensional space", "target variable". Please define these in terms of the oceanographic problem in question.

Reply: We removed these jargons.

Q31. Line 16. "minimizes" – what is varied to do this?

Reply: We revised the description.

Q32. Lines 18-19. "subjecting to the constraint". (A11) looks like a definition of "e" and is not a constraint unless "e" is defined in some other way which needs to be stated.

Reply: We revised the description and re-arranged the equation.

Q33. Line 27. Can there be an explicit expression for '? Where has "b" in (A9) gone to? Table 1. SOM column half way down. "closest" not "closet"!

Reply: We corrected the mistake.

Q34. Figure 3 caption. Please explain "normalized to 2005".

Reply: We revised "normalized to 2005" and in the section 4 added that fCO2 means

trend-removed fCO2 unless specified otherwise.

---

## Author Response (AR1)

Response to referee #1 --------------------------

We thank referee#1 for the thoughtful and constructive feedback on the paper. We have addressed major concerns in the revised manuscript and documented our responses to the referee's comments point-by-point as follows.

1. This Technical note compares the results of three machine learning models for sea surface CO2 mapping. Two of those, self-organizing-maps (SOM) and feedforward neural networks (FNN), have already been used and compared (in the Surface Ocean CO2 Mapping inter comparison initiative, SOCOM) and a new one, the support vector machine (SVM), is introduced in this paper. The SVM performs best but requires big computer memory. This is valuable work as with ever increasing computer power SVM will become available to more users. I have one concern: the resulting model distributions show features that cannot be explained by the CO2 field data. For example there is a CO2 hotspot east of the African coast near the equator where no observations (February) or low CO2 observations (July) are shown in the top panel. In July there is an unexplained hotspot in the Southern Ocean west of South America where there are no observations. I presume these features are produced by the correlation of sea surface CO2 with proxy variables such as SST, SSS CHL and MLD? Are these hotspots known / expected from previous publications? The authors should discuss this further in the discussion of Figure 3.

   Reply: Referee #1 pointed out the CO2 hotspot east of the African coast near the equator and suggested a further discussion. As the issue cannot be cleared without thorough numerical comparisons with outputs of other models, it would go beyond the scope of this manuscript, which is aimed at comparing the three machine learning models. The hotspot CO2 seems quite high comparing to the nearby measurement made by the one cruise in July 1995. It is very difficult to judge whether this is an issue because the observed CO2 shown in the figure is trend removed with a universal rate, which could be another source of uncertainty. Takahashi et al. (2009) shown that the trend could be quite different in different areas. However, it difficult to use multiple rates for global mapping.

2. My final question is: is the dataset produced by SVM available for download somewhere or can it be retrieved from the authors? Could this be added as a supplement possibly?

   Reply: We uploaded the dataset as supplement.

3. Page 1, line 14: include (Goddijn-Murphy et al, 2015).

   Reply: We included the reference.

4. Page 2, line 13: please explain "circular property" and why it can therefore not be used.

Reply: In the manuscript we added "For instance, longitude -180 degree is geographically connected to longitude 180 degree, but numerically they appear to be two extreme longitude values to the models."

5. Page 2, line 14: sine and cosine transformed components of LON and MON? How of MON?

Reply: The transforms are cos(MON*2*pi/12), sin(MON*2*pi/12), cos(Lon*2*pi/360) and sin(LON*2*pi/360). See Zeng et al. (2015). We didn't give the transform here because MON and LON were not used.

6. Page 2, line 14: "The approach" is meaning "Our approach" or "Zeng et al.'s approach"?

Reply: We revised "The approach …" to "Their approach …" to explicitly mean Zeng et al.'s approach.

7. Page 3, line 4: which two CHL products, calculated from OC3 and OCI algorithms?

Reply: The footnote indicates that the products used the OCI algorithm

8. Page 3, line 8, refer to Table 1 here

Reply: We revised "The Appendix summarizes…" to "The Appendix and Table 1 summarize…".

9. Page 3, line 11: 10% of the measurements randomly chosen?

Reply: Yes, they are randomly chosen. We revised "we used 10% of…" to "we randomly chose 10% of…"

10. Page 3, line 12: "dependent of" should be "dependent on".

Reply: We corrected the mistake.

11. Page 3, line 17: insert "" in "all variables "; explain all variables (SST, SSS, CHL, MLD, dSST?).

Reply: We revised the sentence to "we scaled all input variables LAT, SST, SSS, CHL, MLD, and dSST by their minimum and maximum to confine them in the range (0, 1)".

12. Page 4, line 2: give references for preliminary studies.

Reply: We revised "Based on preliminary studies" to "Based on our preliminary correlation analysis".

13. Page4, line 13: replace "to model" with "and modelled".

   Reply We revised the expression accordingly.

14. Page 4, line 18: modeled and observed $CO_2$ of "all / selected/ non-selected" data points?

   Reply: We added "of the selected data points" to the end of the sentence.

15. Page 5, line 6: random 10%?

   Reply: Yes, they were randomly selected. We revised "with 10% of the data" to "with 10% of randomly selected data points".

16. Page 5, line 8: differences are expressed as mean difference standard deviation?

   Reply: Yes.

17. Page 5, line 8: replace "respectively" with "for SOM".

   Reply: We corrected the mistake.

18. Page 5, line 9: give range of measurement uncertainties, how small is small?

   Reply: In the revision we add information for the standard deviation of gridded data for the discussion.

19. Page 5, line 15-17, Fig. 3: The panels for both February and July show features in all three model distributions that are not seen in the field $CO_2$. For example there is a hotspot on the eastern African coast in the western Indian Ocean that is not seen in the observations (top panel). Likewise in July there is an unexplained hotspot west of South America in the Southern Ocean. So, "the models captured the major features of spatial distribution of observed $CO_2$" plus quite a bit more. Can the authors discuss this further in page 5, line 30 - page 6, line 2?

   Reply: See the reply to question 1.

20. Page 8, line 8: "prediction" should be "predictions ".

Reply: We corrected the mistake.

21. Acknowledgements. Include, as suggested on SOCAT's website: "The Surface Ocean CO2 Atlas (SOCAT) is an international effort, endorsed by the International Ocean Carbon Coordination Project (IOCCP), the Surface Ocean Lower Atmosphere Study (SOLAS) and the Integrated Marine Biogeochemistry and Ecosystem Research program (IMBER), to deliver a uniformly quality-controlled surface ocean CO2 database. The many researchers and funding agencies responsible for the collection of data and quality control are thanked for their contributions to SOCAT.

Reply: We revised the acknowledgements as suggested.

22. Table 1:  Add a first column 'Feature', e.g., 1-input space mapping,  2-prediction by, 3-problems, 4-data scaling, 5-results affected by.  Then revise the SVM, FNN, SOM columns accordingly.

Reply: We revised the table as suggested.

23. Table 1, line 9: 'closet' should be 'closest'.

Reply: We corrected the mistake.

24. Figure 3: The labels in white font are too small to read.

Reply: We enlarged the labels.

We thank referee#2 for the thoughtful comments, especially on the appendix. We have revised the appendix substantially to address the reviser's constructive opinions. Here we documented our responses to the reviewer's comments point-by-point.

1. This "technical note" discusses the formation of global maps of surface ocean CO2 from limited measurements using inferred dependence on (latitude, surface temperature SST, salinity, chlorophyll concentration, mixed-layer depth, difference between monthly- and annual-mean SST). The dependence is inferred by three methods: selforganisation map (SOM), feedforward neural network (FNN) and a new method (support vector machine; SVM). The results of these three methods, "trained" on a fraction of the data, are compared with the remaining data. The correlations are not particularly good for any (best at R2 = 0.715 for SVM) considering there are 6 independent variables aiding the fit. However, the results of all three methods for global air-sea CO2 flux are very close and the CO2 maps are visually similar. This similarity extends to a band of high CO2 concentration in February 2005 extending west from Chile where there are apparently no CO2 measurements. This extrapolation from CO2 observations is presumably via a similar feature in (at least) one of the 6 independent variables. There should be more discussion: (i) of the quality of the fit to observed data, especially in relation to the estimates of air-sea flux and the danger that the methods agree with each other more than with reality; (ii) of the extrapolation feature west of Chile (in particular – perhaps also a careful examination for whether there are others) and whether it can be believed in terms of the values of the independent variables – is this set of six values closely approximated somewhere else where there are CO2 measurements constraining the CO2 estimate?

   Reply: First on the quality of fit. As the three model are unbiased, i,e, the mean difference between modeled CO2 and observation is statistically zero, we take the quality here means the correlation and the standard deviation between modeled CO2 and observation. Then quality is not only determined by the capability of the models, but also the variability of the data (we added this information in the revised manuscript).

   Second on the relation to the estimates of air-sea flux. The models may produce differently higher than observed CO2 in some areas and lower CO2 in others. The flux could be largely affect by this and by different wind. That all three models produced similar global fluxes indicates that the effect is small.

   Third on the danger that the methods agree with each other more than with reality. They are quite different issues. The answer to the quality of fit indicates the models cannot agree with reality than the variability of the reality. Whereas, the agreement between models is

determined mainly by their similarity. For example, SVM is considered to be a one layer FNN in some articles; so FNN agrees better with SVM than with SOM.

Fourth on the extrapolation. Yes, the extrapolation approximate unmeasured area with somewhere that has a similar biogeochemical property and CO2 measurement.

2.  Although the organisation and English are generally good, I think some sections and especially the Appendix are unclear/obscure, mainly due to inconsistent or missing explanations, definitions or notation. Most of the following detailed comments are about this aspect.

    Reply: We have revise the appendix substantially to address to issue.

3.  Page 2, lines 12 and 18. "dSST denotes the difference between the monthly and annual means of SST" implies 12 discrete values of dSST; how does this "improve expressing the seasonal variable continuously"?

    Reply: Let's consider three measurements taken on January 1, January 31, and February 1. Using month as the seasonal variable, the variable values of the first two measurements are 1 and the last is 2. However, the seasonally the last two are nearly the same. dSST reflects better the actual change of seawater property caused by season change.

4.  Page 3, Line 13. I think you mean ". . to the range (0, 1) for the SVM . . ."

    Reply: We revised the expression accordingly.

5.  Page 3,. Line 21 (i.e. line after (5)). Why between 0.1 and 0.9 not between 0 and 1? "better" compared with what? Why should scaling the output help?

    Reply: For $f$CO$_2$ close to 0 and 1, and a small change in $f$CO$_2$ requires very large adjustment of model parameters, which slows down the convergence of training. We added this in the revised manuscript.

6.  Page 4, Lines 1-2. "We used Eq. (4) to scale . . SOM". There is no mention of this in Appendix A.1, indeed after (A1) it is stated that the diagonal factors of the scale matrix f are equal to 1.

    Reply: In the appendix section for SOM, we added "In our application, the data for each input

variable were scaled to be unitless by its mean and standard deviation".

7. Page 4, Lines 2-3. "Based on preliminary studies, we applied a factor of 2 to . . SST and CHL . .". What preliminary studies? Is this subjective, i.e. why should SST and CHL be emphasised?

Reply: SOM is indeed subjective. In our knowledge, other applications scaled the data with different subjective factors to change the impact of independent variables on the distance defined by Eq.(A1). In our application, we scaled the data non-subjectively and uses the scale factors to change the impact, which in our opinion is easier to understand. Because CO2 shows a much higher correlation with SST and CHL than with others, we subjectively used a factor of 2 for them. There is no theoretical basis for this choice. We revised the manuscript to address the issue and revised "Based on preliminary studies" to "Based on our preliminary correlation analysis"

8. Page 4, Line 7. "prediction for an input" needs explaining. Inputs are supposed to be known, not "predicted".

Reply: We revised "Making prediction for an input" to "Making a CO2 prediction for an input"

9. Page 4, Lines 8, 9. "map size". In normal language the map size is the earth's surface area. Do you mean resolution, equivalent to the number of CO2 output locations? Please explain / use correct word.

Reply: We revised "the feature map size" to "the number of neuron cells"

10. Page 5, Line 8. "respectively" should be "for SOM"

Reply: We corrected the mistake.

11. Page 5, Lines 11, 15. Please explain "normalized"/"normalization"

Reply: We revised the manuscript and explained "normalized"/"normalization".

12. Page 6. To have value, this needs to be understood in its own terms; the reader should not have to refer to cited references to understand the words used and the overall meaning. Too many words are not defined or explained. Also, it is too abstract. This is a manuscript about

"output" CO2, depending on "inputs" LAT, SST, SSS, CHL, MLD, dSST. Presumably this applies to A.1, A.2 and A.3 – say so and do not use vague terms like "feature space" – at present the reader has to guess what you mean.

Reply: We have revised the appendix substantially according to address the concern.

13. (A.1 . .) Page 6 Line 23. What is "feature space" in oceanographic terms?

Reply: We removed the jargon.

14. Lines 23-24. "usually represented by grid points in two dimensional space". Never mind about "usually"; describe in terms of the problem here.

Reply: We revised the expression to be specific.

15. Line 24. "weight vector w". This name is confusing.

Reply: We changed the symbol and revised the descriptions.

16. page 7, lines 7-8 weights (weight factors) h are defined by (A3). "w" is the result of applying the weights "h" to combine values of "v" at various locations [presumably to represent "v" at grid locations rather than original locations, but this is not clear to the reader. If this the case, then "w" is "gridded v" or "interpolated v"]. See also the comment on page 7 line 21. Line 25. Not "a data vector" which might refer to any vector at all, but "an input data vector" (I guess).

Rely: We changed the symbol and revised the descriptions.

17. page 7, Line 30. "best matching cell (BMC)" needs explaining.

Reply: We revised the descriptions.

18. page 7, Line 30. "minimizing the distance". What is varied to do this?

    Reply: We revised the descriptions.

19. Page 7 Line 4. "matched". Either this is the wrong word or it needs explaining.

    Reply: We revised the descriptions.

20. Line 17. "vector x of input data". In A.1 the input data were "v". Use consistent names for variables.

    Reply: We revised the descriptions.

21. Lines 20-22. You have input data, hidden neurons and output. There should be distinct variable names for each of these, e.g. v, x, y respectively. Here you have y for the hidden neurons and for the output, which is confusing.

    Reply: We changed the symbols and revised the descriptions.

22. Line 21. "w is the weight vector". Indeed this seems correct for its use in (A4) but that is very different from its use in (A1). Use different terms for different quantities (c.f. comment on page 6 line 24).

    Reply: We changed the symbols and revised the descriptions.

23. Line 22. "The training updates the offset and weight parameters". What are the starting values before updating? Do you mean "weight vector" as in line 21?

    Reply: We revised the descriptions. The parameters are initialized randomly between -1 and 1. We added this information in the revised manuscript.

24. Line 23. What is "e" or is it defined by (A5)? Please make this clear.

    Reply: Yes. It is the "e" defined b (A5).

25. Line 24. "modelled . . y" is unclear (especially because you use "y" for hidden neurons and output). Why are two "y" in this line in bold type but not the third or "y" in (A4)?

    Reply: We revised the description. Bold font indicate vector or matrix.

26. Line 24. "w includes both . ." This seems to be defining a vector with more components; it should have a new name.

    Reply: We used a new name.

27. Line 28. " is the learning rate". How is its value decided?

    Reply: The initial value is about 0.25. It is determined by try-and-error. A small value make training slow. A large value make a training diverge. We added the information in the revised manuscript.

28. Line 30. "derivatives of e by w". Do you mean "derivatives of e with respect to w".

    Reply: We revised the description.

29. Page 8, Lines 6-10. "The SVM . . SVM parameters." Is this relevant?

    Reply: We removed this part.

30. Line 14. "independent variables", "high dimensional space", "target variable". Please define these in terms of the oceanographic problem in question.

Reply: We removed these jargons.

31. Line 16. "minimizes" – what is varied to do this?

   Reply: We revised the description.

32. Lines 18-19. "subjecting to the constraint". (A11) looks like a definition of "e" and is not a constraint unless "e" is defined in some other way which needs to be stated.

   Reply: We revised the description and re-arranged the equation.

33. Line 27. Can there be an explicit expression for '? Where has "b" in (A9) gone to? Table 1. SOM column half way down. "closest" not "closet"!

   Reply: We corrected the mistake.

34. Figure 3 caption. Please explain "normalized to 2005".

   Reply: We revised "normalized to 2005" and in the section 4 added that fCO2 means trend-removed fCO2 unless specified otherwise.

Response to referee #3 --------------------------

We thank referee#3 for many valuable comments. As not all questions could be answered satisfactorily without extending the short technical note to a full research paper, the following responses address the referee's opinions in the scope of the technical note.

1. General points: More detail of the exact data application steps are required: Did the application of the methods follow the biogeochemical province-by-province approach of SOCOM, or was all global data combined together?

   Reply: All global data were combined together to trained the models. We did not model the biogeochemical provinces of SOCOM for the reason that not all the provinces have sufficient data for training the models. Dealing with the discontinuity near the borders of provinces are also problematic in global mapping. Although SOCOM compared models by the province-by-province approach, most of the models did not follows the province approach. One of the defined the provinces subjectively, another used SOM to define the provinces, but not of them discussed the border problem in detail.

2. General points: A comment regarding the use of a single trend normalization rate would be welcome. It is known that this is not globally uniform (e.g. Takahashi et al., 2014) and so it would be good to understand the impact of this choice.

   Reply: It's would be interesting to see the impact of using different rates for different areas. However, the approach is a challenge itself as it is difficult to determine the applicable areas for different rates without introducing subjective factors; therefore, it is not realistic for this study that focus on comparing machine learning models.

3. Why are the correlations so much poorer than that achieved by the application of the SOM-FFN approach of Landschutzer et al, 2014)?

   Reply: No model can fit data better than the variability of the data. When CO2 data are subdivided by region or by biogeochemical province, the variability becomes smaller and the data can be fitted better. Landschutzer et al (2014) subdivided the data, so it's not a surprise that their fitting showed better correlations.

4. Within the model validation section, was the random selection of 50% data carried out only once or multiple times? What is the effect of this random selection compared to say, using data clustered around 2005, or only data from regions where pCO2 varies the most, or only

using the most recent data? I would imagine this would be useful information for other researchers looking to apply the methods themselves, whether to map sea surface pCO2 or indeed other biogeochemical parameters. As mentioned above, the study would benefit with comparison with independent dataset e.g. time series at BATS / HOTS. There is very little coverage on uncertainties. More detail on how these are calculated, especially for regions where there are no observational data with which to compare (e.g. South Pacific / Southern Ocean) would be very welcome. This could useful be useful in explaining the anomolous flux feature currently prevalent in Figure 3 in the South Pacific, which is not mentioned in the text and does not appear to be supported by observations or previous studies (e.g. the Takahashi climatology). They are substantial

Reply: The random selection of data was determined by a random number seed. We tested that using different random seeds did change the results significantly. Regarding selecting data clustering around 2005 or recent years, we would like to point out that this may be carried out regionally, but not globally because of scarce measurements. In each month of a year, there might be one or two cruises or none at all doing measurements for the whole globe. Applying the machine learning models to BATS/HOTS should be an independent subject as more data become available the model equation and inputs should be different. For example, the LAT variable should be removed from the model and the measured SST, SSS, and CHL should be used.

5.  Figures: - Figure 2 - unity line is not easily seen. Possibly changing the color of data points to gray could remedy this? - Figure 3 - needs larger labelling as to what they are showing. A column title would be useful, and a more color-blind friendly colorscale.

Reply: We used gray for data points. This improves the figures' appearance.

6.  p5 l7 - what do the uncertainties represent? Are these the standard error of the fit, standard deviation of the mean difference between predicted and observed values? How do these compare to other non neural network methods applied during SOCOM?

Reply: The uncertainty is the standard deviation of the difference between predicted and observed values. We added this to the manuscript. In our opinion, comparing the uncertainties of different models is not meaning full. For example, a model in SOCOM used spline fitting. As we know that spline fitting can fit data perfectly well, but a perfect spline fitting may lead to over interpolation. Another example is SOM. Given a very large number of neuron cells, SOM can also produce perfect fittings, but then the prediction for the spatial distribution of CO2 would be uninterpretable.

7. p5 l9 - what are the measurement uncertainties?

   Reply: The gridded SOCAT includes standard deviation varying from 0.1 μatm to 71.2 μatm. We added this information in the revised manuscript.

8. p5 10 - what is this uncertainty from temperature?

   Reply: Yes, it is. This is not relevant anymore. We used measurements uncertainty for the discussion.

9. p5 11 - what is the average standard deviation of repeat measurements (should also reference)

   Reply: About 12.5 uatm. We added this to the manuscript.

10. p5 13 - why is only july looked at, what is the uncertainty for the full year? How much of this is due to the normalization method?

    Reply: We thought that the manuscript only showed CO2 maps in July and February, so using July as an example was sufficient. Now we included the standard deviation for all months. The effect on the STD by normalization is small. The STD of normalized fCO2 range from 0.1 uatm to 103.1 uatm and the mean is 12.5 uatm; whereas the STD non-normalized fCO2 range from 0.1 uatm to 107.5 uatm and the mean is 14.6

11. p5 25 - there seems some agreement with other studies for 2000 but substantial disagreement with other estimates (Wanninkhof et al., 2013, Rodenbeck et al., 2015) for 2010. This is surprising given that this is when there are most observational data and so it could be assumed that this era would be best modelled. Equally it is rather worrying that the same models as used in the SOCOM study are showing substantially higher estimates for the air-sea CO2 flux for the same input dataset. Is this related to the choice of wind field or how the mapped pCO2 fields are built? How do the mapped pCO2 fields compare with other methods? Some comment on this discrepancy would be greatly appreciated. In particular, comment on how fluxes for years other then 2000 are calculated would be useful as this is not currently explained. Is the systematic trend of 1.5uatm/year simply reintroduced.

    Reply: Yes, the flux estimate is highly dependent on wind products as shown by Wanninkhof et al. (2013) and Zeng et al. (2014). We added a short comment to the manuscript.

12. p5 l27 - the within-model differences are smaller, but this would be expected as they are essentially iterations of a similar technique. More disconcerting is the substantial offset of this group of models with other independent approaches. As mentioned above, more comment/discussion on this aspect would be useful.

    Reply: SOCOM shows that FNN agree well with other models in general. Inter-comparison of model by different authors is important but beyond the scope of this manuscript.

[revised manuscript text omitted]

---

## Author Response (AR2)

We would like to thank the topic editor for his/her comments. We hope the following responses and the revised contents in the manuscript sufficiently addressed the comments.

1. All three referees commented on the models producing features (especially a "hotspot in the Southern Ocean west of South America" – referee 1) where there are no fCO2 data. You really should respond to these comments in the final text – the comments will be public and readers will be able to judge your responses. I think you can look at the independent (input) variables: do these extrapolated fCO2 features correspond to features in one or more input variables? Especially important for confidence in the models: is the set of input variable values closely approximated somewhere else so that the model features where there is not fCO2 data are in fact constrained by fitted fCO2 data?

   Response: We added a paragraph in section 7 to discussed the hotspot issue. The hotspots in the Southern Ocean west of South America can also be seen in the modeled $CO_2$ of Takahashi et al. (2014). For comparing with their results, we changed the color scheme of figure 3 to the same color scheme. At this stage, it is not realistic to precisely answer the question "is the set of input variable values closely approximated somewhere else…", because this requires removing input variables one-by-one and retraining all models multiple times. Tracking "somewhere else" is not possible for FNN. It is possible for SVM and SOM, but requires revising their source code. In section 5, we added a paragraph to plain that the SOM and SVM do not have over extrapolation risk. Since the hotspots appeared in all models, the risk of accepting them would not be high.

2. Page 2 equation (1). Referee 3 commented on your use of a single value of trend for all locations. You have partly answered this at the end of section 5 but you should relate the statements there to the referee's question. "normalisation" does not make a clear link.

   Response: We added "The trend in Eq.(2) cannot be modelled directly by the models. One approach to deal with the problem is to normalize the measurements to a reference year using a global rate and only model the nonlinear component." in section 4 to make "normalisation" more clear.

3. Page 2 equation (2) and dSST. I suspect that dSST is not described properly in the text. "difference between the monthly and annual means of SST" is not continuous, it means 12 discrete values, same values on January 1 and January 31 and different values on January 31 and February 1. But your response says you are avoiding this problem.

   Response: You are right. The dSST is not continuous in time. While month changes mechanically, dSST reflect better the effect of season on seawater property. For example, the season does

affect the equator as much as the Northern and Southern oceans. We revised the last sentence in section 2 to address this.

4.  Section 4. Perhaps there should he sub-headings to make clearer that page 3 lines 20-25 are about SVM, page 4 lines 1-15 are about FNN and page 4 lines 16-25 are about SOM.

    Response: We added sub-headings as suggested.

5.  Appendix A3. I think the kernel function definition (A15) should come directly after A(9) where φ is introduced.

    Response: We rearranged the equation as suggested.

6.  Equations (A12) and (A14). Somehow "c" seems to have been replaced by "α" but the relation between them is not stated.

    Response: We are very grateful that this comment makes us to re-examine equations in the appendix. Derivations for the equations are very long and complicated in textbooks. In trying to use a minimal number of equations to deliver the concepts of the models, we made mistakes, especially for SVM. Now we added the missing links between "c" and "α". We avoided using the dot product operator for vector all together. It can be a trap for mistake.